# REWARD INFLATION PARADIGM THROUGH THE LENS OF MONETARY ECONOMICS

## ABSTRACT

Reward is fundamental to reinforcement learning (RL), where the agent treats it as an incentive to maximize. This appearance is akin to a rational human who maximizes their income. However, in the real economy, money expands to stimulate economic growth. Inspired by this principle of monetary economics, we introduce a novel RL paradigm, **reward inflation**, which gradually increases the reward scale during training. Analogous to inflationary policies used by central banks to stimulate economic growth, reward inflation acts as an incentive stimulus for agents to accelerate policy learning. Reward inflation can be applied in two ways: fixed or adaptive. Motivated by the Fed's monetary policy, we propose **FedeRL**, a dynamic controller for adaptive inflation. Theoretical analysis suggests that the effect of reward inflation is threefold: (1) induces recency bias in temporal-difference learning, (2) amplifies policy gradients, and (3) enhances neural activation. Empirical results corroborate these insights, showing that moderate inflation improves performance on continuous control tasks. Moreover, FedeRL performed even better than fixed inflation and outperformed comparable baselines. By translating economic growth principles into RL, our approach offers a novel perspective that strengthens policy optimization and addresses fundamental RL objectives. The implementation code will be made publicly available.

## 1 INTRODUCTION

Reinforcement Learning (RL) has shown promising results in diverse domains, such as robotics (Tang et al., 2025), games (Souchleris et al., 2023), art (Lee et al., 2023), autonomous driving (Zhao et al., 2024), recommendation systems (Afsar et al., 2022), finance (Bai et al., 2025), and recently gained traction to improve large language models (LLMs) (Havrilla et al., 2024).

Since RL maximizes the sum of rewards from its behaviors instead of minimizing errors from data, rewards play a central role in its trial-based learning. Recent works have explored various techniques to enhance reward design (Eschmann, 2021), including reward normalization (Schaul et al., 2021), intrinsic motivation (Aubret et al., 2023), and curriculum learning (Gupta et al., 2022). These approaches provide options for improvement by suggesting the task-specific structure or semantics of the reward for given tasks. However, the temporal dynamics of the reward scale throughout training remain underexplored.

In this work, we present a novel perspective on RL by drawing upon principles of economic growth and monetary economics, proposing the concept of **reward inflation**. To realize this, we introduce nominal reward in RL, analogous to the issuance of fiat money in monetary economics which lacks intrinsic value. Although fiat money holds no intrinsic value, it acts as a powerful tool for economic growth because it incentivizes people to create real value and allows central banks to regulate its supply. In this vein, we view the nominal reward as a driving incentive to maximize the real episode reward, proposing a new framework that accelerates learning by modulating the magnitude of nominal rewards over time, mirroring the regulation of the money supply.

While the introduction of nominal rewards opens up various possibilities, this study specifically focuses on the positive effects of inflation within an economy. Central banks typically increase the money supply in a gradual and consistent manner while carefully avoiding economic instability, because, despite the resulting burden of inflation, it injects vitality into the economy and

ultimately drives growth. Drawing on this expansionary monetary policy, we implement reward inflation through a simple mechanism of steadily increasing the magnitude of nominal rewards. At a more granular level, an increased money supply in real economies typically induces money illusion, higher liquidity, and reduced unemployment in the short run. Mirroring these phenomena, our theoretical analysis suggests that reward inflation leads to recency bias, amplifies gradient magnitudes, and reduces neuron deactivation rates. These insights are corroborated by our empirical results.

Reward inflation is mainly determined by the inflation rate. While it is feasible to use fixed rewards at an appropriate level, we propose **FedeRL** as an adaptive controller for reward inflation. FedeRL dynamically adjusts the inflation rate in response to learning progress, using policy gradient norms as a proxy for training momentum. Drawing inspiration from the Federal Reserve's monetary policy and the Taylor Rule, FedeRL reduces inflation when gradient norms spike to stabilize training and increases it when progress slows.

Our empirical study centers on MuJoCo locomotion tasks (Todorov et al., 2012) with SAC (Haarnoja et al., 2018). Experiments show that the moderate fixed inflation (e.g., about 2% per 100K steps) improved performance compared to non-inflation baseline results. Among various inflation rates, a set of best inflation rates improved performance by an average 7.79% over non-inflation baseline results. However, an excessive inflation (10% per 100K steps) impaired performance by an average of 2.96%, demonstrating the importance of the moderate rate of inflation. FedeRL further outperformed fixed inflation results, with an average 9.27% improvement over non-inflation baseline results. Compared with alternative strategies such as learning rate scheduling, prioritized experience replay, and reward or critic normalization, FedeRL consistently demonstrates superior results. We further demonstrate the effectiveness of reward inflation on Box2D tasks (SAC) and the Arcade Learning Environment (ALE) (Bellemare et al., 2013) with DQN (Mnih et al., 2015).

Our contributions are summarized as follows:

- ○ We introduce a novel paradigm, the reward inflation, linking with monetary economics, as a straightforward methodology to accelerate RL.
- ○ We propose FedeRL, a novel inflation rate adjustment mechanism within our paradigm, analogous to a central bank's monetary policy.
- ○ We provide theoretical analyses that suggest three effects of reward inflation: recency bias, gradient norms, and network activation, which respectively correspond to money illusion, liquidity, and unemployment in the real economy.
- ○ We empirically validate our approach through experiments, showing consistent improvements over baselines and comparable methods.

## 2  RELATED WORKS

Reward design serves as a key determinant of learning dynamics. Early work focused on stabilizing value estimates across magnitudes, such as potential-based reward shaping (Ng et al., 1999), automatic rescaling (Van Hasselt et al., 2016), return-based normalization (Schaul et al., 2021), and handling temporal heterogeneity in reward structures (Dann & Thangarajah, 2021). More recent approaches aim to automate reward construction, such as reward shaping as a meta-learning problem (Zou et al., 2019), intrinsic reward shaping (Yuan et al., 2023; Aubret et al., 2019), and a bi-level optimization framework (Hu et al., 2020). Curriculum RL also involves reward shaping, such as sparse-to-dense rewards (Park et al., 2025; Freitag et al., 2024), gradual sub-goals (Anca et al., 2023), and reward machine-based approach (Shukla et al., 2023).

Our paradigm shares the goal of improving learning efficiency but differs in that we preserve reward semantics and instead modulate their temporal influence through gradual scaling. A closely related family is normalization, such as reward normalization (Henderson et al., 2018), return normalization (Schulman et al., 2017), and value normalization (PopArt) (Van Hasselt et al., 2016; Hessel et al., 2019). These methods stabilize signal scale and benefit multi-task learning but introduce non-stationarity and do not guarantee performance gains. We employ normalization approaches as comparative methods given their simplicity and widespread adoption.

Instead of altering rewards, some methods focused on memory utilization to optimize the learning signal indirectly. Prioritized experience replay (PER) assigns importance based on TD error (Schaul et al., 2015; Zha et al., 2019; Zhang et al., 2020), with extensions considering state distribution (Sun

et al., 2020), freshness (Ma et al., 2022), and learnability (Sujit et al., 2023). Other approaches learn sampling policies (Zha et al., 2019) or decouple actor-critic sampling (Saglam et al., 2023). Recently, (Yamani et al., 2025) prioritized transitions via reward prediction errors using an enhanced critic. While our approach relates to memory in that it affects the effective weight of past experiences, it does not require altering the underlying memory or sampling mechanisms.

## 3 METHODOLOGY

In this section, we provide standard RL formulations, followed by the notation and definition of the reward inflation paradigm. Based on this paradigm, we introduce FedeRL to dynamically control the inflation rate.

### 3.1 PRELIMINARIES

An RL problem is defined as a Markov Decision Process (MDP) as $\mathcal{M} = \mathcal{S}, \mathcal{A}, \mathcal{P}, \mathcal{R}, \gamma$, where $\mathcal{S}$ and $\mathcal{A}$ denote the state and action spaces, $\mathcal{P}$ is the transition function, $\mathcal{R}$ the reward function, and $\gamma$ the discount factor. A policy $\pi \in \Pi$ is a mapping from states to action distributions, where $\Pi$ denotes the set of all possible policies. At each timestep $t$, the agent samples an action $a_t \in \mathcal{A}$ as $a_t \sim \pi(\cdot \mid s_t)$, where $s_t \in \mathcal{S}$. Given the current state $s_t$ and action $a_t$, the next state is sampled as $s_{t+1} \sim \mathcal{P}(\cdot \mid s_t, a_t)$, and the agent receives a reward $r_t \in \mathcal{R}$. The return at timestep $t$ is defined as the sum of future discounted rewards $G_t = \sum_{k=t}^{T} \gamma^{k-t} r_k$, where $T$ denotes the end of the episode. The objective of RL is to find the optimal policy $\pi^*$ which maximizes the expected episode return starting from the initial state as $\pi^* = \arg\max_\pi \mathbb{E}_\pi [G_0]$.

### 3.2 REWARD INFLATION

We denote *global step* $g \in \mathbb{N}_0$—a unique index that counts the total number of environment interactions throughout training—as a timeline of inflation. The inflation is represented as time-varying *inflation rate* or *annual inflation rate* $\rho_Y(g\,;Y)$, where $Y$ denotes the number of global steps per year, and $\rho_Y(g\,;Y) > -1$. From the definition, *per-step inflation rate* $\tilde{\rho}(g)$ can be derived as

$$\tilde{\rho}(g) := (1 + \rho_Y(g\,;Y))^{\frac{1}{Y}} - 1. \tag{1}$$

For example, $\rho_Y(2026 \cdot 10^5\,; 10^5) = 0.02$ represents a 2% annual inflation at $g = 2026 \cdot 10^5$, where one year corresponds to $10^5$ global steps. In this case, the per-step inflation rate is $\tilde{\rho}(2026 \cdot 10^5) = 1.02^{1/10^5} \approx 0.00000198$. Then we define *inflation level* $I(g) > 0$ at global step $g$ as

$$I(g) := \begin{cases} 1, & \text{if } g = 0 \\ \prod_{k=1}^{g} (1 + \tilde{\rho}(k)), & \text{if } g > 0 \end{cases} \tag{2}$$

which is multiplied to the *real reward* $r_t$ to produce the current *nominal reward* $r_t'$ as

$$r_t'(g) := r_t \cdot I(g). \tag{3}$$

At any given $g$, the inflation level $I(g)$ indicates the degree of inflation relative to the initial level $I(0)$, where the nominal rewards are identical to the real rewards.

Note that reward inflation is applied prior to insertion into the replay buffer, ensuring that the stored rewards remain fixed. With reward inflation, RL uses the nominal reward $r_t'$ instead of the real reward $r_t$. Therefore, reward inflation alters the original MDP by introducing an evolving reward function that varies linearly from $\mathcal{R}$ to $\mathcal{R}'$ with $g$, such that $\mathcal{M}'(g) = \{\mathcal{S}, \mathcal{A}, \mathcal{P}, \mathcal{R}'(g), \gamma\}$. RL with this new MDP maximizes the *nominal return* $G_t' = \sum_{k=t}^{T} \gamma^{k-t} r_k'$, instead of *real return* $G_t$.

As training progresses, the inflation level grows exponentially if $\tilde{\rho} > 0$, a condition termed *positive inflation*. If $\tilde{\rho} = 0$, the inflation remains constant, reducing the paradigm to standard RL when held throughout training. When $-1 < \tilde{\rho} < 0$, rewards contract over time. This *negative inflation* (or deflation) yields effects opposite to positive inflation. We exclude the unnatural case of $\tilde{\rho} \leq -1$, where rewards vanish or reverse in order, distorting the original problem. With inflation, the reward scale is non-stationary. Still, the optimality remains the same since only the scale of $\mathcal{R}$ is changing, as a linear transformation to $\mathcal{R}$ does not change the optimal policy (Ng et al., 1999).

### 3.3 REWARD INFLATION AND MONETARY ECONOMICS

To bring more economic insights, we draw an analogy between reward inflation and the growth of the money supply. Consider nominal rewards be a currency, denoted as "RWD", with an initial exchange rate of 1 RWD = 1 (real) reward. As households or firms seek to maximize their utility or revenue, an RL agent seeks to maximize total RWDs in a single episode. When the money supply expands, both prices and incomes rise, which defines inflation. Likewise, under reward inflation, RWD devalues (e.g., 1.2 RWD = 1 reward). According to *money neutrality*, these changes are only nominal and should not affect real values (productivity in the real economy and performance in RL).

However, nominal changes can, in fact, affect the real economy for several reasons, and one of them is adjustment delays. When people see their income increase, they initially believe their purchasing power has improved and spend more, even though prices also rise and real purchasing power stays constant. This temporary misperception, known as *money illusion*, can stimulate economic activity until expectations catch up with the new price level. Similarly, reward inflation can prompt an RL agent to reinforce certain new actions more strongly than under no inflation, thereby encouraging broader and faster policy changes before the agent fully adjusts to the new reward scale.

Money supply growth also provides *liquidity*, encouraging investment and reducing unemployment until inflation is fully reflected. Analogously, increased nominal rewards stimulate training, strengthen gradients, and finally can activate more neurons than in no inflation. However, excessive money supply can cause the economy to become inefficient and unstable. Likewise, overly large reward inflation can confuse the agent and degrade performance. The following sections reconfirm the above analogies in theoretical and empirical analysis.

### 3.4 FEDERL: CONTROLLING INFLATION RATES

Inflation rates can either be fixed or adaptive. We introduce FedeRL that adaptively adjusts inflation rates during training. Just as the Fed stabilizes economic growth by lowering interest rates during slowdowns and raising them during overheating, FedeRL adapts the inflation rate based on training dynamics. Specifically, it increases the rate to promote learning and reduces it to mitigate instability, thereby ensuring stable policy improvement.

To diagnose slowdown or overheating, FedeRL monitors the gradient norm $\mathcal{G}(\phi) = \|\nabla_\phi J(\phi)\|_2$ of the policy $\pi_\phi$, where $J(\phi)$ is the policy loss. An exponential moving average (EMA) tracks its trend:

$$m_k(x) = \beta \cdot m_{k-1}(x) + (1 - \beta) \cdot x_k, \tag{4}$$

with $x$ being the target sequence and $k$ the update step. FedeRL maintains short- and long-term EMAs, $m_k^s(\mathcal{G}(\phi))$ and $m_k^l(\mathcal{G}(\phi))$, with decay rates $\beta_s < \beta_l$. Their ratio,

$$m_k^{s/l}(\mathcal{G}(\phi)) = m_k^s(\mathcal{G}(\phi)) \, / \, m_k^l(\mathcal{G}(\phi)), \tag{5}$$

serves as an indicator: $m_k^{s/l} > 1$ suggests overheating, while $m_k^{s/l} < 1$ indicates slowdown.

To control inflation, FedeRL adjusts the rate using:

$$\rho_Y^{new} = \rho_Y^* - \lambda_1(m_k^{s/l} - 1) - \lambda_2(m_k^{s/l} - m_{k-1}^{s/l}), \tag{6}$$

where $\rho_Y^*$ is the natural inflation rate, and $\lambda_1, \lambda_2$ denote sensitivity to level and change, respectively. This update encourages $m_k^{s/l}$ to converge to 1, stabilizing training at $\rho_Y^*$. It derives from the Taylor Rule, a widely used reference for interest rate adjustment. Since equation 6 may cause abrupt rate changes, we additionally apply a soft update to determine the next $\rho_Y$ as follows:

$$\rho_Y = \tau_\rho \cdot \rho_Y^{old} + (1 - \tau_\rho) \cdot \rho_Y^{new}, \tag{7}$$

where $\rho_Y^{\text{old}}$ is the previous rate, and $\tau_\rho$ the soft update coefficient. FedeRL updates $m_k^{s/l}$ and $\rho_Y$ $K$ times per year.

## 4 THEORETICAL ANALYSIS OF REWARD INFLATION

In this section, we analyze the primary effects of reward inflation. We begin with the typical case of positive inflation, followed by negative inflation. All proofs are provided in Appendix B.

## 4.1 ANALYSIS ON POSITIVE INFLATION RATE

The positive inflation rate continuously expands the nominal rewards at a constant rate. In this setting, inflation grows monotonically, thereby making the following two lemmas naturally hold.

**Lemma 1** (Expected nominal reward proportional to inflation level). *Let the real rewards $r_t$ be i.i.d. across global steps. Then, the expected current nominal reward is proportional to the current inflation level $I(g)$ as*

$$\mathbb{E}\left[r'_t(g)\right] \propto I(g). \tag{8}$$

**Lemma 2** (Expected nominal reward gap proportional to inflation gap). *Suppose that the real rewards $r_t$ are i.i.d. across global steps. Then, for any global steps $p < q$ with inflation gap $I(q) - I(p)$, the expected nominal rewards satisfy*

$$\mathbb{E}\left[r'_t(q)\right] - \mathbb{E}\left[r'_t(p)\right] \propto I(q) - I(p). \tag{9}$$

As nominal rewards depend on inflation level $I(g)$, the inflation-reflected true action-value function $Q^\pi$ also grows with $I(g)$, assuming the inflation is stable in a single episode.

**Proposition 1** ($Q^\pi$ proportional to inflation level). *Assume that real rewards $r_t \in \mathbb{R}$ are i.i.d. across global steps, and that the inflation level $I(g)$ is approximately constant within an episode. Then the true action-value function $Q^\pi$ for the policy $\pi$ is approximately proportional to the current $I(g)$:*

$$Q^\pi(s, a) \propto I(g). \tag{10}$$

The $Q$-learning deals with double trouble: the original $Q$-estimation and the inflation adaptation.

**Assumption 1** (Inflation adaptation gap for inflation-tracking $Q_\theta$). *Let the Q-network $Q_\theta$ be trained via temporal-difference (TD) updates to track the inflation-reflected action-value function $Q^\pi$.*

*After $n$ TD updates under inflation level $I(g)$, the deviation is expressed as*

$$Q^\pi(s, a) - Q_\theta^{(n)}(s, a) = \varepsilon_{\text{approx}}(s, a) \cdot \Delta_n(g), \tag{11}$$

*where $\varepsilon_{\text{approx}}(s, a)$ denotes the approximation error, and $\Delta_n(g)$ the inflation adaptation gap. Since the replay buffer contains transitions with mixed inflation levels, the adaptation gap does not vanish even as $n \to \infty$, and instead admits a nonzero lower bound that grows with inflation level $I(g)$:*

$$\liminf_{n \to \infty} \|\Delta_n(g)\|_2 \geq c \cdot I(g), \quad c > 0. \tag{12}$$

Among the components of a transition, only the reward is affected by inflation. Assuming transitions are i.i.d., states and actions are independent of the global step $g$. Thus, the expected value of $Q^\theta(s, a)$ remains unchanged over $g$.

**Assumption 2** (Approximate expected action-value stationarity). *Let $s_t(g)$ and $a_t(g)$ denote the state and action acquired at global step $g$. We assume that between two global steps $p < q$, the learned action-value expectations remain approximately invariant by the principle of indifference:*

$$\mathbb{E}\left[Q_\theta(s_t(p), a_t(p))\right] \approx \mathbb{E}\left[Q_\theta(s_t(q), a_t(q))\right], \tag{13}$$

*since both $(s_t(p), a_t(p))$ and $(s_t(q), a_t(q))$ are invariant to the global step $g$.*

We now derive that the TD target $y_g$ at global step $g$ also scales with the inflation level $I(g)$, similar to the nominal rewards and the $Q$-function.

**Lemma 3** (Expected TD target proportional to inflation level). *Let the real rewards $r_t$ are i.i.d. across global steps. Then, the expected TD target $y_g$ is proportional to the current $I(g)$ as*

$$\mathbb{E}\left[y_g\right] \propto I(g). \tag{14}$$

Thereafter, the expected TD target gap between two different global steps follows the inflation gap.

**Lemma 4** (Expected TD target gap). *Suppose that transitions used for TD targets $y$ are i.i.d.. Then, for any global steps $p < q$, the expected TD targets satisfy*

$$\mathbb{E}[y_t(q)] - \mathbb{E}[y_t(p)] \propto I(q) - I(p). \tag{15}$$

From the above lemmas, we formalize the two effects of positive reward inflation: **recency bias** and **gradient norm amplification**.

**Theorem 1** (Recency bias induced by positive reward inflation). *Reward inflation with $\tilde{\rho} \geq 0$ induces a time bias in $Q_\theta$ and $\pi_\phi$, causing the agent to more sensitive to newer experiences.*

**Theorem 2** (Gradient norm amplification by positive reward inflation). *Reward inflation with $\tilde{\rho} > 0$ amplifies the expected gradient norms of both $Q_\theta$ and $\pi_\phi$, approximately proportional to the current inflation level $I(g)$ as*

$$\mathbb{E}\left[\|\nabla_\theta L(\theta)\|_2\right], \ \mathbb{E}\left[\|\nabla_\phi J(\phi)\|_2\right] \propto I(g). \tag{16}$$

**Remark 1** (Money illusion versus money neutrality). As in Assumption 1, under positive inflation, agents must update $Q_\theta$ to catch up with $Q^\pi$, compensating for outdated inflation effects. However, $Q_\theta$ cannot distinguish whether changes in nominal rewards come from real improvements or inflation, leading to **money illusion**—the agent maximizes nominal return, mistaking it for real value. This causes increased sensitivity to recent transitions, as shown in Theorem 1.

Theorem 2 illustrates how reward inflation propagates to the gradient level. The amplified gradient norm acts as **liquidity provision**, as discussed in 3.3. Since it reflects the total training signal per update, its increase corresponds to greater monetary supply for neuronal adjustment. These observations motivate the following conjecture.

**Conjecture 1** (Stimulative neural effect of reward inflation). *Positive reward inflation reduces deactivated neurons, and the extent of the reduction increases proportionally with the inflation rate.*

### 4.2 ANALYSIS ON NEGATIVE INFLATION RATE

We now analyze the effect of negative inflation (deflation), which can be seen as the reverse case of the two theorems under positive inflation.

**Theorem 3** (Reversed recency bias induced by reward deflation). *Reward deflation with $-1 < \tilde{\rho} < 0$ induces a time bias in $Q_\theta$ and $\pi_\phi$, causing the agent to more sensitive to older experiences.*

Since the agent tends to rely more on past experiences under reward deflation, it may decelerate the innovation process in RL. However, it can also be useful when training overheats and needs to be slowed down.

**Theorem 4** (Gradient norm attenuation by reward deflation). *Reward deflation with $-1 < \tilde{\rho} < 0$ attenuates the expected gradient norms of both $Q_\theta$ and $\pi_\phi$, approximately proportional to the current inflation level $I(g)$ as*

$$\mathbb{E}\left[\|\nabla_\theta L(\theta)\|_2\right], \ \mathbb{E}\left[\|\nabla_\phi J(\phi)\|_2\right] \propto I(g). \tag{17}$$

Except for the inflation level $I(g)$ decreases towards 0 during the deflation, the proportional relationship remains the same as Theorem 2.

In reward deflation, the effect of money illusion still holds, but with reversed direction. While Theorem 2 corresponds to liquidity provision, Theorem 4 corresponds to **liquidity tightening**, thus proposing the following conjecture, as a reversed version of Conjecture 1.

**Conjecture 2** (Restrictive neural effect of reward inflation). *Reward deflation increases the number of deactivated neurons, and the extent of increase is proportional to the deflation rate.*

We theoretically analyzed the effects of reward inflation on recency bias, gradient norm, and left Conjectures on neuron activation, which we empirically examine in the next section.

## 5 EXPERIMENT

In this section, we present the experimental results to address the following key questions:

- How do the reward inflation mechanism and FedeRL work in practice?
- Does empirical analysis confirm theoretical claims and economic principles?
- Does the reward inflation policy, including FedeRL, achieve competitive performance against the no-inflation baseline and other methods?

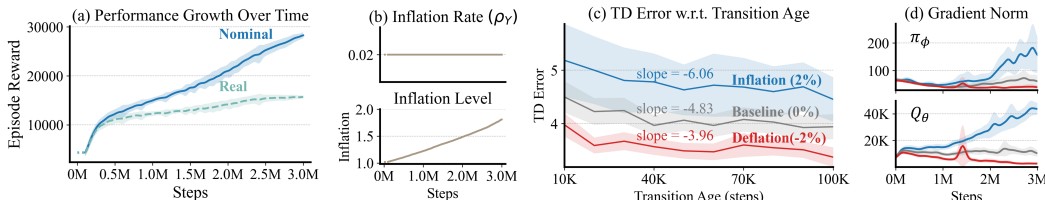

Figure 1: (a) The nominal and real performance growth over training time under 2% inflation. (b) The inflation rate and inflation level $I(g)$ over the training time. (c) The TD error w.r.t. the transition age in replay buffer, across the inflation rates. (d) The changes of gradient norms during training.

## 5.1 EXPERIMENTAL SETUP

We conducted experiments on eight representative tasks from the MuJoCo benchmark (Todorov et al., 2012). These tasks have been widely used for evaluation in RL domains, for their powerful generality and moderate difficulty for current mainstreams. As the backbone RL algorithm, we used Soft Actor-Critic (SAC) (Haarnoja et al., 2018) due to its strong empirical performance and broad applicability across continuous control domains. We further conducted additional experiments on Box2D tasks (SAC) and the Arcade Learning Environment (ALE) (Bellemare et al., 2013) with DQN (Mnih et al., 2015). All experiments are averaged over 10 seeds. For the annual inflation rate $\rho_Y(g \; ; \; Y)$, we set $Y = 10^5$ steps as one year. For FedeRL, through naive search, we set $\beta_s = 0$ (instantaneous update), $\beta_l = 0.83$, $K = 12$, $\rho_Y^* = 0.02$, $\lambda_1 = 0.2$, $\lambda_2 = 0.2$, and $\tau_\rho = 0.83$ as general settings. More detailed hyperparameters are provided in Appendix C.1.

## 5.2 EMPIRICAL CHARACTERISTICS OF REWARD INFLATION AND FEDERL

**Nominal and real episode reward**  Maximizing nominal return is one distinctive property of reward inflation. Figure 1(a,b) illustrates the gap between nominal and real episode reward, alongside the change in the inflation level $I(g)$ during training. This example uses `HalfCheetah-v4` with 2% inflation rate. The nominal episode reward grew consistently faster than the real one, in proportion to $I(g)$. Note that we report the real episode reward as the performance metric.

**Recency bias and gradient norm**  To examine how reward inflation affects recency bias and gradient norm, we compared agents under 2% (inflation), 0% (baseline), and −2% (deflation). Recency bias was measured from the dependence of TD error on transition age. Figure 1(c) shows that older transitions, having been sampled more frequently, generally exhibit lower TD errors. However, inflation steepened the TD error gap between old and new transitions, while deflation flattened it. This result supports Theorems 1 and 3. Figure 1(d) shows the evolution of gradient norm over time, corresponding to Theorems 2 and 4. These patterns were consistently observed in all tasks.

**Effects on neuron activation**  The two Conjectures in our theoretical analysis suggested that reward inflation may reduce the number of deactivated neurons, whereas deflation may increase them. To examine this empirically, we measured the number of dead neurons (He et al., 2015) in both the $Q_\theta$ and $\pi^\phi$ networks during training. A dead neuron is defined as one that consistently outputs zero, contributing nothing to learning. Dead neurons are initially raised with ReLU activation; however, a similar phenomenon is generally observed with other activation functions (Gustineli, 2022). Figure 2(a) shows how the proportion of dead neurons evolved across various inflation rates.

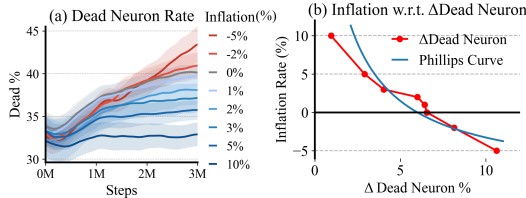

Figure 2: (a) The changes in dead neuron rate during training from deflation (-5%) to inflation (10%). (b) The total increased rate of dead neurons under each inflation rate.

Table 1: Performance comparison under varying inflation rates (upper rows) and other methods (lower rows) across MuJoCo locomotion tasks. The dagger (†) marks the best value among fixed inflation rates. The last column reports the average normalized score across tasks.

| Method | HalfCheetah | Walker2d | Humanoid | Swimmer | Hopper | Ant | Pusher | Reacher | NS |
|---|---|---|---|---|---|---|---|---|---|
| -2% | 14020±1082 | 4342±461 | 5300±176 | 92±43 | 2421±460 | 3929±1090 | -24.25±4.10 | -3.46±0.23 | 61.7 |
| 0% | 15241±734 | 4399±527 | 5826±313 | 116±36 | 2869±299 | 4763±1642 | -23.28±2.89 | -3.41±0.23† | 81.9 |
| 1% | **15707**±715† | 5007±560 | 5859±363 | 121±25 | 2821±406 | 5245±1312† | -22.71±2.87 | -3.42±0.23 | 88.2 |
| 2% | 15607±372 | 5102±520† | 5861±199 | 136±11† | 3040±499 | 4346±1267 | -22.07±1.58† | -3.42±0.24 | 91.3† |
| 3% | 15091±927 | 4690±624 | 5873±306† | 126±11 | 3035±445 | 4392±1298 | -22.58±1.58 | -3.44±0.24 | 83.4 |
| 5% | 14484±996 | 4615±518 | 5545±492 | 131±12 | 3161±346 | 4795±1089 | -22.20±0.89 | -3.45±0.23 | 82.8 |
| 10% | 14186±1200 | 4167±650 | 5209±1241 | 129±19 | **3166**±288† | 4248±1219 | -24.60±1.47 | -3.54±0.33 | 63.0 |
| FedeRL | 15621±991 | **5395**±452 | **6048**±436 | **143**±13 | 3105±293 | 5180±1352 | **-21.99**±1.23 | -3.43±0.24 | **95.4** |
| 2% (lr) | 14566±1006 | 4922±515 | 5417±298 | 77±41 | 2899±359 | 4458±1888 | -23.95±2.88 | -3.42±0.24 | 61.2 |
| PER | 14734±1934 | 4797±592 | 4542±2290 | 57±23 | 2683±529 | **5762**±979 | -26.41±2.98 | -3.43±0.24 | 86.7 |
| RN | 15022±436 | 2272±1009 | 520±107 | 111±18 | 648±70 | 187±112 | -26.49±5.52 | **-3.40**±0.10 | 30.6 |
| RTN | 11877±369 | 4454±571 | 4451±633 | 87±21 | 2742±463 | 4046±1603 | -23.41±2.05 | -3.45±0.13 | 57.6 |

Surprisingly, a clear trend emerged: higher inflation rates consistently reduced the number of dead neurons, while deflation increased them relative to the baseline. This suggests that stronger total signal input may suppress neuron deactivation, aligning with our Conjectures. Figure 2(b) plots the increase in dead neuron rates at the training end versus inflation rates. Interestingly, the relationship resembles the Phillips Curve (Phillips, 1958), reflecting a trade-off between inflation and unemployment, and echoes Keynesian ideas of activating idle resources (Samuelson & Solow, 1960).

However, fewer deactivated neurons do not necessarily imply better performance. Although the dead neuron rate was lowest under 10% inflation, this setting yielded the poorest results (see 5.3 for details). Fewer dead neurons indicate greater equality among neurons, but not necessarily higher effectiveness: while such equality may broaden participation, it can reduce overall efficiency. This neuron activation issue reflects a broader trade-off between capacity utilization (Sokar et al., 2023) and the benefits of sparsity (Dufort-Labbé et al., 2024), and remains an active research area. Therefore, gradual reward inflation may enhance capacity utilization, potentially benefiting RL networks.

**Operation of FedeRL** FedeRL adaptively adjusts inflation in response to changes in the short-long gradient ratio $m^{s/l}$. Figure 3 shows how FedeRL controls inflation on two contrasting cases. In HalfCheetah-v4, $m^{s/l}$ surged rapidly at the beginning of training, indicating overheating from strong learning signals. As this task is known for early and rapid performance gains, FedeRL responded by sharply lowering the inflation rate, even entering deflation. Once $m^{s/l}$ stabilized, the inflation rate recovered to the preset natural rate of 2% and remained relatively steady thereafter. In contrast, Swimmer-v4 showed the opposite pattern. Due to its slow, reward-scarce early phase,

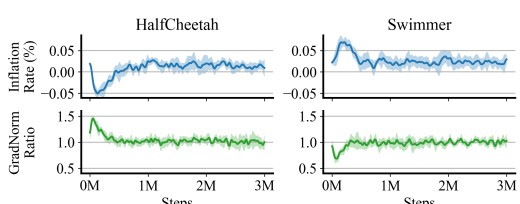

Figure 3: Inflation rates (top) adjusted by FedeRL in response to the observed short-long gradient norm ratios $m^{s/l}$ (bottom), on HalfCheetah-v4 and Swimmer-v4.

$m^{s/l}$ initially declined, prompting FedeRL to raise the inflation rate to stimulate learning. As training progressed and $m^{s/l}$ stabilized, the inflation rate gradually converged to the natural level.

### 5.3 PERFORMANCE COMPARISON OF INFLATION RATES AND OTHER APPROACHES

**MuJoCo locomotion tasks** To evaluate reward inflation, we ran eight MuJoCo locomotion tasks using inflation rates of −2%, 0%, 1%, 2%, 3%, 5%, and 10%, as well as FedeRL. We also compared with learning-rate scheduling (±2%), PER, RN, and RTN. PopArt was omitted due to severe degradation and instability.

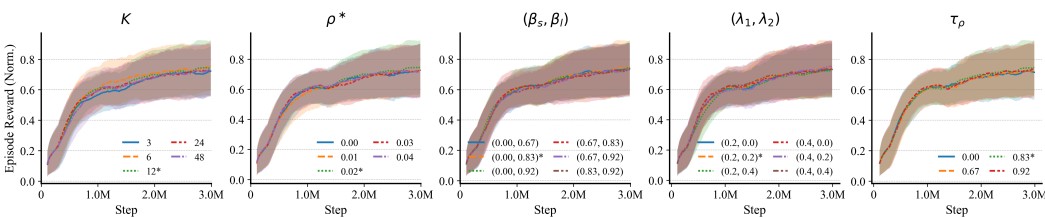

Figure 4: Learning curves of FedeRL varying 5 sets of hyperparameters. Default values are marked with $*$. Episode rewards are normalized across 4 tasks.

Table 1 shows results averaged over the final 300K steps. The last column reports normalized scores (NS). Overall, except for one task, both 1% and 2% inflation outperformed the 0% baseline, while deflation ($-2\%$) consistently reduced performance. As shown in Figure 1(c), deflation dampens the distinctiveness of new transitions. Although optimal rates varied per task, 2% achieved the highest NS, forming an inverted U-shaped trend. The 10% rate obtained the best score in `Hopper-v4` but worst NS overall with large variance, indicating hyperinflation risk. On average, 2% improved results by 4.79%, the best rates by 9.27%, while 10% and $-2\%$ degraded results by 2.96% and 9.72%.

Meanwhile, FedeRL has more flexibility in determining the inflation rate. Notably, FedeRL reached the highest NS across all methods, showing its effectiveness in controlling inflation to improve performance. Even compared to the best fixed inflation rates for each task, FedeRL outperformed them in half the tasks. In terms of general performance, it outperformed the no-inflation baseline by 9.27% and the 2% inflation by 4.45%.

While reward inflation amplifies gradient norms, learning rate (lr) inflation reflects the gradient more strongly during updates. Despite apparent similarities, lr inflation operates independently of the gradient, making them orthogonal methods with differing effects. Empirically, 2% reward inflation outperformed the baseline, whereas 2% lr inflation underperformed, confirming their distinct effects. PER relates to TD error magnitude but underperformed the baseline except on `Ant-v4`. Reward inflation applies a global temporal slope on TD errors, whereas PER emphasizes only large ones, sometimes causing bias. RN and RTN stabilize reward scales but may distort semantics and showed inconsistent performance. In contrast, modest reward inflation and FedeRL produced stable and consistent gains.

### 5.4 FURTHER ANALYSIS

**Box2D environment** We additionally conducted experiments on two Box2d environments with no-inflation baseline, 2% inflation, and FedeRL with SAC. Figure A.1 illustrates the learning curves. These tasks are relatively simple, and thus, the baseline is enough to reach almost the highest scores. However, agents with reward inflation still slightly outperformed the baseline agent.

**Arcade Learning Environment** Extending our empirical study to the discrete domain, we evaluated the efficacy of reward inflation on 30 ALE Atari 2600 games using the DQN algorithm with a 2% inflation rate. Table 2 summarizes the average performance over the last 300K steps, with the final row showing the average Human-Normalized Score. The results indicate that 2% inflation

Table 2: Summary of performance comparison on 30 ALE tasks.

| Metric | 0% | 2% |
|---|---|---|
| # Winning Tasks | 4 | **26** |
| Mean HNS | 176.6 | **192.0** |

led to performance improvements in most games, with an average gain of 8.72%. Detailed results and learning curves are illustrated in Table A.3 and Figure A.2.

**Hyperparameter study of FedeRL** We conducted a study to investigate the impact of FedeRL hyperparameters on performance. They can be summarized as $Y, \rho_Y^*, K, \beta_s, \beta_l, \lambda_1, \lambda_2$, and $\tau_\rho$. Here, the annual step count $Y$ exists primarily for interpretability and can be eliminated; the essence lies in how the step inflation rate $\tilde{\rho}$ is determined independent of $Y$. Therefore, in this study, we searched for the target inflation rate $\rho_Y^*$ while fixing $Y = 10^5$. The remaining hyperparameters, while di-

verse, ultimately serve as variables to control the volatility of inflation updates. The inflation rate fluctuates more aggressively as $K$, $\lambda_1$, and $\lambda_2$ increase, and as $\tau_\rho$ decreases. For $\beta_s$ and $\beta_l$, smaller values with larger gap between them lead to more aggressive fluctuations in gradient norm measurements, thereby inducing larger variations in the inflation rate. Figure A.3 displays the mean absolute deviation curves of the inflation rate to demonstrate the volatility under different hyperparameter conditions.

Figure 4 compares the normalized performance across four MuJoCo tasks by varying five specific factors ($K$, $\rho^*$, $(\lambda_1, \lambda_2)$, $(\beta_s, \beta_l)$, $\tau_\rho$) while fixing others to their default values. Default values are marked with an $*$ in the labels of each graph. For $\beta_s$, $\beta_l$, and $\tau_\rho$, 0.00 denotes a hard update, while 0.67, 0.83, and 0.92 correspond to effective sample sizes of 3, 6, and 12, respactively. Although slight advantages were observed at $K = 6, 12$ and $\rho^* = 0.2$, overall hyperparameter variations did not exhibit significant performance differences or distinct trends. This can be interpreted as the result of inflation being applied gradually, meaning that differences in inflation volatility do not exert a substantial impact. In summary, while FedeRL involves numerous hyperparameters, it can be considered a method that is reasonably robust to these parameters.

**Learning rate inflation** Although Table 1 includes only a 2% learning rate (lr) inflation, examining a broader range of inflation rates allows for a more objective comparison. The left column of Table A.4 presents the results of lr inflation ranging from 0% to 4% across four MuJoCo tasks. Applying inflation to the learning rate failed to demonstrate significantly better results compared to the 0% baseline. While reward inflation may appear similar to learning rate inflation at first glance, these results demonstrate that they clearly differ in empirical outcomes. Conversely, the right column of Table A.4 shows that lr deflation down to -4%, which corresponds to exponential learning rate decay, resulted in overall improvements. This can be attributed to the inherent benefits of learning rate decay. Although learning rate decay is an independent and non-mutually exclusive method with distinct mechanisms and objectives compared to reward inflation, making a direct comparison somewhat imperfect, we included it in this analysis because both methods share the common characteristic of modulating signals over the course of training.

**Environmental uncertainty** Although previous experiments were conducted in noise-free environments, real-world settings often involve various forms of uncertainty. To assess whether reward inflation undermines robustness, we injected three types of Gaussian noise, (1) observation ($0.01\sigma$), (2) reward ($1.0\sigma$), and (3) action ($0.1\sigma$), into four MuJoCo tasks and compared SAC agents under 0% and 2% inflation. As shown in Figure A.4, both agents degrade under noisy conditions, but the 2% agent retains comparable or slightly better performance than the baseline. This suggests that the benefits of reward inflation may persist to some extent even in the presence of noise.

## 6 DISCUSSION

We proposed a novel RL paradigm, reward inflation, inspired by economic principles from monetary theory. Theoretical and empirical analyses revealed that modest inflation can alleviate optimization bottlenecks in RL by amplifying learning signals, improving adaptation, and reducing neuron inactivity. To further improve training stability, we introduced FedeRL, an adaptive inflation controller based on gradient-norm dynamics, which outperformed both baseline and fixed inflation strategies. Our results show that modest reward inflation and FedeRL significantly influence learning dynamics and policy performance, improving over a no-inflation baseline and related approaches. These findings highlight the utility of macroeconomically inspired reward shaping for better temporal credit assignment and training stability in RL.

A primary limitation is that the exponential growth of inflation makes it numerically unsustainable in the long run. To address this in long-horizon tasks, future iterations must introduce concepts like currency reform or mechanisms to decay the inflation rate to zero. We also acknowledge that FedeRL, as an application of this paradigm, is limited by its heuristic design and high number of hyperparameters. Future large-scale analyses will be essential to solidify the theoretical foundation and refine the proposed methodology. Finally, while this paper utilized monetary economics to offer a simplified interpretation, the analogy between economic growth and RL suggests a rich avenue for future exploration. We leave the integration of more complex economic concepts and the development of varied applications as promising directions for future research.

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

## A  LLM USAGE

We used an LLM solely as a general-purpose assistant for polishing grammar and wording. It did not contribute to research ideation, experiments, or writing content.

## B  PROOFS

### B.1  PROOF OF LEMMA 1

*Proof.* The definition of $r'_t$ is

$$r'_t = r_t \cdot I(g). \tag{18}$$

Therefore,

$$\begin{aligned} \mathbb{E}\left[r'_t\right] &= \mathbb{E}\left[r_t \cdot I(g)\right] \\ &= \mathbb{E}\left[r_t\right] \cdot I(g). \end{aligned} \tag{19}$$

$$\therefore \mathbb{E}\left[r'_t\right] \propto I(g) \quad \square \tag{20}$$

### B.2  PROOF OF LEMMA 2

*Proof.*

$$\begin{aligned} \mathbb{E}\left[r'_t(q)\right] - \mathbb{E}\left[r'_t(p)\right] &= \mathbb{E}\left[r_t \cdot I(q)\right] - \mathbb{E}\left[r_t \cdot I(p)\right] \\ &= \mathbb{E}\left[r_t\right] \cdot I(q) - \mathbb{E}\left[r_t\right] \cdot I(p) \\ &= \mathbb{E}\left[r_t\right] \cdot (I(q) - I(p)). \end{aligned} \tag{21}$$

$$\therefore \mathbb{E}\left[r'_t(q)\right] - \mathbb{E}\left[r'_t(p)\right] \propto I(q) - I(p) \quad \square \tag{22}$$

### B.3  PROOF OF PROPOSITION 1

*Proof.* The true action-value function $Q^\pi$ using nominal rewards $r'_t$ is defined as

$$\begin{aligned} Q^\pi(s, a) &:= \mathbb{E}\left[\sum_{t=0}^{T} \gamma^t r'_t(g)\right] \\ &= \mathbb{E}\left[\sum_{t=0}^{T} \gamma^t r_t \cdot I(g)\right] \\ &\approx \mathbb{E}\left[\sum_{t=0}^{T} \gamma^t r_t\right] \cdot I(g), \end{aligned} \tag{23}$$

assuming that the variation of inflation level $I(g)$ is trivial in a single episode and can thus be approximated as constant. Thus, the $Q^\pi$ is approximately proportional to the current inflation level $I(g)$. $\square$

### B.4  PROOF OF LEMMA 3

*Proof.* The TD target $y_t(g)$ is defined as

$$y_t(g) = r'_t(g) + \gamma Q_\theta(s_{t+1}, \pi(s_{t+1})). \tag{24}$$

Based on the Lemma 2 and Proposition 1, each term in the right-hand side is proportional to the current inflation level $I(g)$. Therefore, the claim holds. $\square$

### B.5 PROOF OF LEMMA 4

*Proof.* Let $d_t(g) = \{s_t(g), a_t(g), r'_t(g), s_{t+1}(g+1)\}$ be a transitions acquired after global step $g$ and to be sampled i.i.d. from the replay buffer. The $Q$-network $Q_\theta$ is trained using TD targets $y_t(q)$ and $y_t(p)$ defined as

$$
\begin{aligned}
y_t(q) &= r'_t(q) + \gamma Q_\theta(s_{t+1}(q+1), \ \pi(s_{t+1}(q+1))), \\
y_t(p) &= r'_t(p) + \gamma Q_\theta(s_{t+1}(p+1), \ \pi(s_{t+1}(p+1))).
\end{aligned}
\tag{25}
$$

To compare the expectations $\mathbb{E}[y_q]$ and $\mathbb{E}[y_p]$, we analyze each term separately.

By Lemma 3, the first terms satisfy

$$
\mathbb{E}[r'_t(q)] - \mathbb{E}[r'_t(p)] \propto I(q) - I(p).
\tag{26}
$$

For the second terms, their expected difference is near-zero by Assumption 2 as

$$
\underbrace{\mathbb{E}\left[Q_\theta(s_{t+1}(q+1), \ \pi(s_{t+1}(q+1)))\right]}_{\text{same by indifference}}
\tag{27}
$$

$$
-\underbrace{\mathbb{E}\left[Q_\theta(s_{t+1}(p+1)), \ \pi(s_{t+1}(p+1)))\right]}_{\text{same by indifference}} \approx 0.
\tag{28}
$$

Combining the gap in the first terms with near-zero gap in the second term yields the desired proportionality. $\qquad\square$

### B.6 PROOF OF THEOREM 1

*Proof.* Let $d_t(g) = \{s_t(g), a_t(g), r'_t(g), s_{t+1}(g+1)\}$ be a transitions acquired after global step $g$ and to be sampled i.i.d. from the replay buffer. For any global steps $p < q$, during a stochastic-gradient step the $Q$-network minimizes

$$
\begin{aligned}
L(\theta) &= \tfrac{1}{2}\left(y_t(q) - Q_\theta(s_t(q), a_t(q))\right)^2 \\
&\quad + \tfrac{1}{2}\left(y_t(p) - Q_\theta(s_t(p), a_t(p))\right)^2.
\end{aligned}
\tag{29}
$$

The parameter update is

$$
\begin{aligned}
\theta \leftarrow \theta + \eta[&\delta_t(q)\nabla_\theta Q_\theta(s_t(q), a_t(q)) \\
&+ \delta_t(p)\nabla_\theta Q_\theta(s_t(p), a_t(p))],
\end{aligned}
\tag{30}
$$

$$
\delta_t(g) := y_t - Q_\theta(s_t(g), a_t(g)),
\tag{31}
$$

where $\delta_t(g)$ denotes TD error and $\eta$ is the step size.

Here, the expected TD error gap is

$$
\begin{aligned}
&\mathbb{E}\left[\delta_t(q)\right] - \mathbb{E}\left[\delta_t(p)\right] \\
&= \mathbb{E}\left[y_t(q)\right] - \mathbb{E}\left[y_t(p)\right] \\
&\quad -\gamma\left(\mathbb{E}\left[Q_\theta(s_t(q), a_t(q))\right] - \mathbb{E}\left[Q_\theta(s_t(p), a_t(p))\right]\right).
\end{aligned}
\tag{32}
$$

Lemma 5 and Assumption 2 each show

$$
\mathbb{E}\left[y_t(q)\right] - \mathbb{E}\left[y_t(p)\right] \propto I(q) - I(p),
\tag{33}
$$

$$
\mathbb{E}\left[Q_\theta(s_t(p), a_t(p))\right] - \mathbb{E}\left[Q_\theta(s_t(q), a_t(q))\right] \approx 0,
\tag{34}
$$

respectively. It follows that the magnitudes of the TD error is larger for the more recent transition $d_t(q)$ than $d_t(p)$, proportional to the inflation gap $I(q) - I(p)$, highlighting the recency bias on $Q$-learning.

Subsequently, the policy gradient is described as

$$
\nabla_\phi J(\phi) = -\mathbb{E}_{(s,a)\sim\pi_\phi}\left[\nabla_\phi \log \pi_\phi(a \mid s) \cdot Q_\theta(s,a)\right].
\tag{35}
$$

As the policy depends on $Q_\theta(s,a)$ for corresponding transitions, the recency bias observed in $Q$-learning is directly inherited by the policy update. $\qquad\square$

### B.7 Proof of Theorem 2

*Proof.* The loss for $Q_\theta$ and its gradient are given by

$$L(\theta) = \mathbb{E}\left[(Q_\theta(s,a) - y)^2\right], \tag{36}$$

$$\nabla_\theta L(\theta) = \mathbb{E}\left[2 \cdot (Q_\theta(s,a) - y) \cdot \nabla_\theta Q_\theta(s,a)\right]. \tag{37}$$

Since the TD target $y$ scales with the inflation level $I(g)$, the TD error $|Q_\theta(s,a) - y|$ also grows proportionally to $I(g)$, under Assumption 1 that $Q_\theta$ does not immediately track the inflated target. As a result, both the loss $L(\theta)$ and the gradient norm $\|\nabla_\theta L(\theta)\|_2$ grow approximately in proportion to $I(g)$.

Next, the loss for $\pi_\phi$ and its gradient are given by

$$J(\phi) = -\mathbb{E}_{(s,a) \sim \pi_\phi}\left[\log \pi_\phi(a \mid s) \cdot Q_\theta(s,a)\right], \tag{38}$$

$$\nabla_\phi J(\phi) = -\mathbb{E}_{(s,a) \sim \pi_\phi}\left[\nabla_\phi \log \pi_\phi(a \mid s) \cdot Q_\theta(s,a)\right]. \tag{39}$$

According to Proposition 1, the $Q$-values scale proportionally with $I(g)$ in expectation. Since the policy gradient is a linear function of $Q_\theta(s,a)$, the gradient norm $\|\nabla_\phi J(\phi)\|_2$ also scales proportionally with $I(g)$. $\qquad \square$

### B.8 Proof of Theorem 3

*Proof.* The condition $-1 < \tilde\rho < 0$ reverses the relationship between $p$ and $q$ in Lemma 1 as

$$I(p) - I(q) > 0. \tag{40}$$

This subsequently reverses the Lemma 3 as

$$\mathbb{E}\left[r'_t(p)\right] - \mathbb{E}\left[r'_t(q)\right] \propto I(p) - I(q). \tag{41}$$

From the equation equation 32 in the Theorem 2, the expected TD error gap can be rewritten as

$$\mathbb{E}\left[\delta_t(p)\right] - \mathbb{E}\left[\delta_t(q)\right]$$
$$= \mathbb{E}\left[y_t(p)\right] - \mathbb{E}\left[y_t(q)\right]$$
$$-\gamma\left(\mathbb{E}\left[Q_\theta(s_t(p), a_t(p))\right] - \mathbb{E}\left[Q_\theta(s_t(q), a_t(q))\right]\right). \tag{42}$$

The equation equation 41 and Assumption 2 each show

$$\mathbb{E}\left[y_t(p)\right] - \mathbb{E}\left[y_t(q)\right] \propto I(p) - I(q), \tag{43}$$

$$\mathbb{E}\left[Q_\theta(s_t(p), a_t(p))\right] - \mathbb{E}\left[Q_\theta(s_t(q), a_t(q))\right] \approx 0. \tag{44}$$

It follows that the magnitudes of the TD error is larger for the older transition $d_t(p)$ than $d_t(q)$, proportional to the deflation gap $I(p) - I(q)$, highlighting the reversed recency bias on $Q$-learning.

Subsequently, the policy depends on $Q_\theta(s,a)$ for corresponding transitions, the reversed recency bias observed in $Q$-learning is directly inherited by the policy update. $\qquad \square$

### B.9 Proof of Theorem 4

*Proof.* Since the TD target $y$ scales with the inflation level $I(g)$, in the condition $-1 < \tilde\rho < 0$, the TD error $|Q_\theta(s,a) - y|$ contracts proportionally to $I(g)$, under Assumption 1 that $Q_\theta$ does not immediately track the inflated target. As a result, both the loss $L(\theta)$ and the gradient norm $\|\nabla_\theta L(\theta)\|_2$ contract approximately in proportion to $I(g)$.

According to Proposition 1, the $Q$-values scale proportionally with $I(g)$ in expectation. Since the policy gradient is a linear function of $Q_\theta(s,a)$, the gradient norm $\|\nabla_\phi J(\phi)\|_2$ also contracts proportionally with $I(g)$. $\qquad \square$

Table A.1: SAC hyperparameter configuration.

| Parameter | Value |
|---|---|
| Total timesteps | 3M |
| Replay buffer size | 1M |
| Target smoothing coef. ($\tau$) | 0.005 |
| Batch size | 256 |
| Warm-up steps | 100K |
| Learning rate ($Q_\theta$) | 0.001 |
| Learning rate ($\pi_\phi$) | 0.0003 |
| Update interval ($Q_\theta$) | 1 |
| Update interval ($\pi_\phi$) | 2 |
| Update interval (target $Q_\theta$) | 1 |
| Initial entropy coef. ($\alpha$) | 1.0 |
| Automatic $\alpha$ tuning | True |
| Activation | ReLU |
| Optimizer | Adam |

## C  EXPERIMENTAL DETAILS

### C.1  RL HYPERPARAMETERS

If not specified, all experiments share the SAC hyperparameters listed in Table A.1.

These hyperparameters mostly follow those of the implementation of CleanRL (Huang et al., 2022). Note that 3M for total timesteps is 3 times more than the common use-case in MuJoCo benchmark, as we intended to investigate the long-term effect of reward inflation compared to the baseline. Also, we used 20 times more warm-up steps (100K) than in the original implementation (5K) for a more stable result, as 5K warm-up steps occasionally induced severely poor performance for both the baseline agent and ours.

We adjusted the discount factor $\gamma$ for *Hopper-v4* and *Ant-v4* to reduce performance volatility, as in Table A.2. Note that the baseline agent with those hyperparameters showed better performance on

Table A.2: Discount factor ($\gamma$) specification.

| Task | Discount factor ($\gamma$) |
|---|---|
| Hopper-v4 | 0.995 |
| Ant-v4 | 0.975 |
| Others | 0.99 (default) |

each task than the default setting ($\gamma = 0.99$).

For DQN agents, we used the same setting from CleanRL implemented for ALE.

### C.2  RESOURCES

All experiments were conducted on a system with multiple Linux servers including GPUs as NVIDIA RTX 4090, RTX A6000, RTX A5000, and TITAN RTX, paired with CPUs as AMD EPYC 9354 32-Core, AMD EPYC 9554 64-Core, AMD Ryzen Threadripper PRO 5975WX 32-Cores and AMD Ryzen Threadripper 3970X 32-Cores, respectively.

## D  ADDITIONAL RESULTS

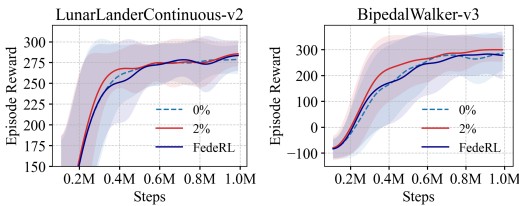

Figure A.1: Learning curves of the baseline, 2% inflation, and FedeRL on Box2d tasks.

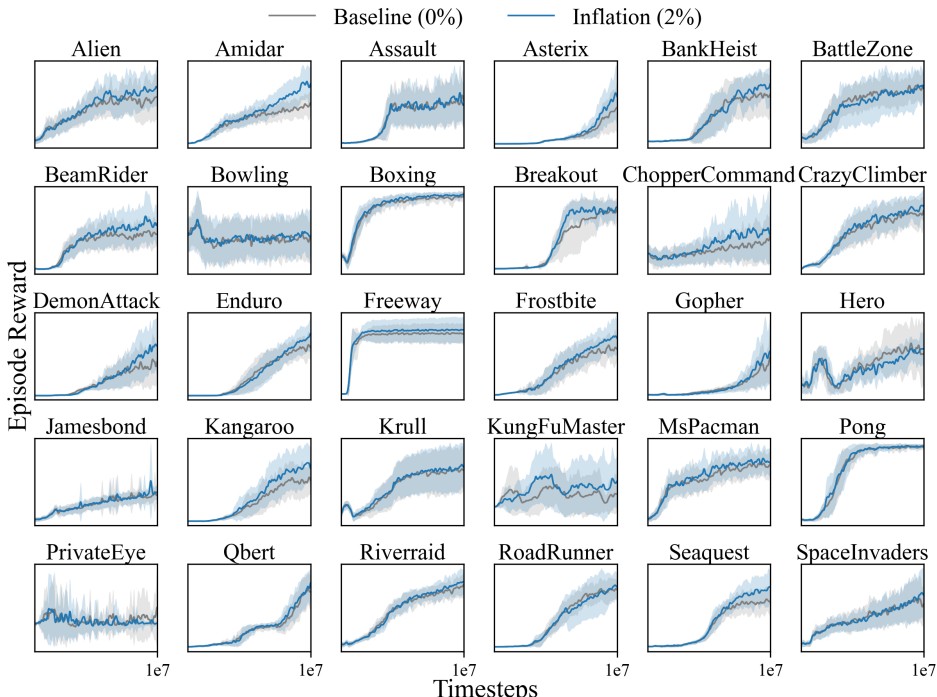

Figure A.2: All learning curves of DQN agents in ALE games under reward inflation (0% vs 2%).

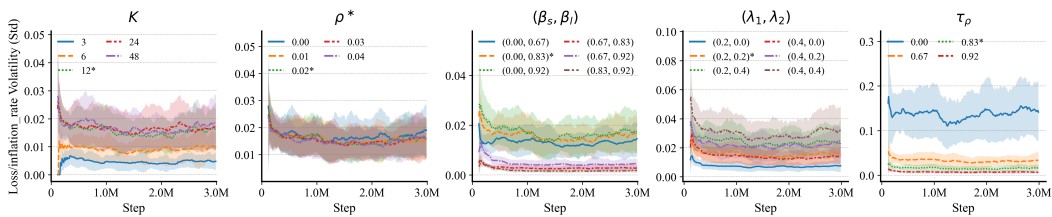

Figure A.3: Mean absolute deviation curves of inflation rate under various FedeRL hyperparameter settings. Note that target inflation $\rho^*$ does not impact volatility.

Table A.3: Performance comparison of DQN agents in ALE games under reward inflation (0% vs 2%).

| Task | 0% | 2% | Task | 0% | 2% |
|---|---|---|---|---|---|
| Alien | 1172±83 | **1317±115** | Frostbite | 2173±262 | **2455±326** |
| Amidar | 238±13 | **310±34** | Gopher | 5474±1416 | **6680±2823** |
| Assault | 3313±266 | **3383±259** | Hero | **2366±1080** | 1997±323 |
| Asterix | 9764±1798 | **13050±1786** | Jamesbond | 410±27 | **422±36** |
| BankHeist | 509±219 | **547±174** | Kangaroo | 6379±2206 | **8434±2753** |
| BattleZone | **19591±584** | 18569±6034 | Krull | 6393±2208 | **6605±2280** |
| BeamRider | 6421±542 | **7658±705** | KungFuMaster | 4978±5458 | **7937±7738** |
| Bowling | 21.99±0.14 | **22.44±0.37** | MsPacman | 1990±113 | **2148±113** |
| Boxing | 84±2 | **86±3** | Pong | 19±1 | **19±1** |
| Breakout | 325±65 | **370±7** | PrivateEye | **265±319** | 145±116 |
| ChopperCommand | 980±364 | **1270±479** | Qbert | 5763±619 | **6176±516** |
| CrazyClimber | 80573±6658 | **86253±8914** | Riverraid | 7031±328 | **7272±396** |
| DemonAttack | 24941±3557 | **33136±2855** | RoadRunner | **26251±1803** | 25485±6279 |
| Enduro | 1012±218 | **1085±258** | Seaquest | 3320±359 | **3906±423** |
| Freeway | 23±3 | **25±4** | SpaceInvaders | 1108±87 | **1118±62** |

Table A.4: Performance comparison under varying learning rate inflation (left) and deflation (right) rates.

| | LR Inflation | | | | | LR Deflation (Decay) | | | |
|---|---|---|---|---|---|---|---|---|---|
| Rate | HalfCheetah | Walker2d | Humanoid | Swimmer | Rate | HalfCheetah | Walker2d | Humanoid | Swimmer |
| 0% | **15241**±734 | 4399±527 | **5826±313** | **116±36** | 0% | 15241±734 | 4399±527 | **5826±313** | 116±36 |
| 1% | 14860±1560 | 4729±422 | 5634±151 | 107±35 | -1% | 15366±575 | 4617±508 | 5752±322 | 93±42 |
| 2% | 14566±1006 | **4922±515** | 5417±298 | 77±41 | -2% | 15181±1087 | 4982±498 | 5728±234 | 124±24 |
| 3% | 14643±895 | 4646±660 | 5282±226 | 90±41 | -3% | **15799±583** | **5167±630** | 5667±202 | 127±27 |
| 4% | 13854±905 | 4502±507 | 5302±280 | 113±30 | -4% | 15033±1438 | 4925±479 | 5787±168 | **132±15** |

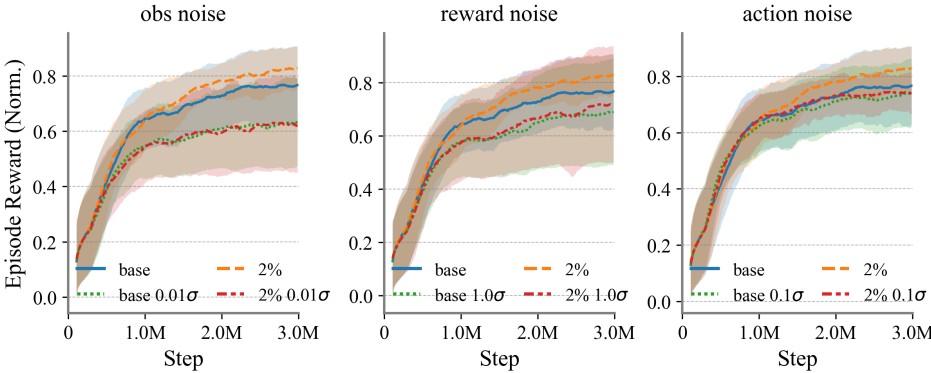

Figure A.4: Non-inflation (0%) vs. inflation (2%) on more uncertain conditions.

