# OpenReview forum: "Reward Inflation Paradigm Through the Lens of Monetary Economics"
_ICLR.cc/2026/Conference — Submitted to ICLR 2026_

### Official Review · Reviewer_2CBW · 2025-10-29

**Soundness:** 2
**Presentation:** 2
**Contribution:** 2
**Rating:** 4
**Confidence:** 3

**Summary:**

This paper proposes FedeRL, a method inspired by monetary economics that dynamically scales reward functions.
Specifically, the authors first motivate why adjusting the scales of the rewards can be beneficial during training, using concepts from macroeconomics.
This is followed by the FedeRL algorithm, which, inspired by how central banks work, adjusts the inflation rates based on gradient norms of the function approximator.
Finally, the authors show empirically that FedeRL can reduce dead neurons and improve performance in MuJoCo when used with the soft actor-critic algorithm.
Overall, the paper proposes an RL algorithm that achieves improved empirical results, but its theoretical framing is underdeveloped and, in its current form, does not add meaningful insight.

**Strengths:**

The proposed method is simple to implement and appears to perform well on MuJoCo tasks.
The idea of modifying the rewards to stabilize training is an interesting one, which may provide new insights into algorithm designs.

**Weaknesses:**

The connection to economic theory appears largely superficial.
While the paper draws some analogies to economic concepts, these do not seem to inform the design of the proposed algorithm (FedeRL) in a meaningful way.
The “economic inspiration” is mentioned primarily at a narrative level rather than as a source of substantive methodological insight.
As a result, the framing may give the impression of novelty without contributing actual conceptual or theoretical value to the RL problem.

The proposed method, while effective, also introduces eight new hyperparameters ($Y$, $\beta_s$, $\beta_l$, $K$, $\rho_Y^{*}$, $\lambda_1$, $\lambda_2$, $\tau_\rho$), and a study of these parameters is lacking.

**Questions:**

1. Can the authors clarify precisely how the economic analogy contributes to the algorithm’s design?
2. How sensitive is the algorithm regarding the hyperparameters?
3. In the implementation of SAC, are the rewards scaled first before being stored into the replay buffer? Or is it scaled on-the-fly during training depending on the global step? If I understand correctly, Assumption 1 is based on the setting where all rewards are scaled by the same factor. However, if the replay buffer contains rewards that are scaled differently, it is not clear to me what Q-learning would converge to, if it converges at all.

---

> ### Author Response · Authors · 2025-12-03
>
> We thank the reviewer for recognizing the intuitive nature and novel insights of our method.
>
> We also sincerely appreciate the valuable feedback that encouraged us to deeply reflect on the essence of our work.
>
> **Weakness & Question 1. Superficial analogy and weak methodological insight**
>
> We fully understand the concern that drawing analogies from macroeconomics may seem superficial and potentially tangential to RL.
>
> Our initial strategy was to utilize the economic concepts primarily for motivation and interpretation, while maintaining overall focus on more engineering parts.
>
> However, upon careful review and reconsideration based on the reviewer’s comment, we recognized that the economic narrative in the **Introduction** appeared somewhat fragmented, which weakened the connection between the economic and engineering aspects.
>
> We have therefore restructured the **Introduction** to clarify the broader economic narrative, and slightly revised **Section 3.3** to address the more granular and mechanical details.
>
> The connection between the economic and engineering aspects is actually deeper than mere analogy, as it provided the foundational insight for our core mechanism.
>
> The analogy is crucial because when attempting to explain why a seemingly burdensome reward distortion (inflation) results in superior performance, the concepts of economic growth and monetary policy offer the most effective and congruent explanation.
>
> While many dislike inflation, a continuous, moderate inflation is an intended result of monetary policy, as it stimulates economic actors, encourages faster adoption of new behaviors, and ultimately accelerates economic growth.
>
> In the RL context, this translates to the expectation that inflation, despite being a distortion, might lead to faster adoption of new actions, quicker adaptation to new states, and ultimately accelerating the convergence to the optimal policy. (Here, we deliberately avoided deeper dives into concepts like "credit" or "interest rates" to prevent the paper from becoming unmanageably large and to maintain its focus on the primary RL contribution, and we appreciate your understanding of this concern regarding academic scope.)
>
> To answer the **Question 1**:
>
> The algorithm design for reward inflation itself (previous to FedeRL) was contributed by the economic analogy connecting economic growth and the role of currency/incentives.
>
> Then, the FedeRL algorithm directly mimics the Taylor Rule, one of the canonical mechanisms for adjusting interest rates based on economic conditions.
>
> Therefore, FedeRL’s design was almost directly derived from the analogy: "interest rate adjustment based on economic conditions" ↔ "reward inflation rate adjustment based on learning conditions."
>
> **Weakness & Question 2. FedeRL hyperparameter study**
>
> Like other reviewers, the reivewer pointed out the lack of a comprehensive hyperparameter study for FedeRL. We fully agree that this was an omission, and we acknowledge that appropriate tuning guides were neglected.
>
> We have addressed this by performing a detailed hyperparameter search study and reflecting the findings in the revision (please see newly added **Section 5.4, "Hyperparameter Study of FedeRL"**).
>
> While our analysis identified slightly recommended values, the overarching conclusion remains that FedeRL exhibits overall low sensitivity to its hyperparameters, which suggests it can be robustly applied without extensive tuning.
>
> **Question 3. Q-learning under mixed scaled rewards**
>
> This crucial question regarding the non-stationarity created in the replay buffer has been clarified in the revision. The reward inflation is applied before the experience is stored in the replay buffer, meaning the reward scale in the buffer is fixed at the inflation level corresponding to the storage time.
>
> However, as stated in **Proposition 1**, the true Q-value is proportional to the real-time inflation level, which changes constantly.
>
> **Assumption 1** posits that the Q-net approximates this True Q, but a necessary error, the Inflation adaptation gap, remains due to the scale difference among rewards in the buffer.
>
> Therefore, Q-learning under reward inflation theoretically chases an ideal true Q that reflects the real-time inflation level, and even a perfect approximation retains the inflation adaptation gap caused by the reward scale variation.
>
> As noted in the response to Reviewer #Qmnv (Q1), this intentional distortion (the scale difference) is precisely what drives the effectiveness of reward inflation by creating the desired recency bias.

---

### Official Review · Reviewer_Qmnv · 2025-10-30

**Soundness:** 3
**Presentation:** 3
**Contribution:** 3
**Rating:** 4
**Confidence:** 4

**Summary:**

This paper introduce a reward inflation method that increase the reward scale during training.  Theoretical analysis suggests three main effects: (1) recency bias, (2) gradient norm amplification, and (3) enhanced neural activation.

**Strengths:**

1. This idea is interesting and novel
2. the result is promising

**Weaknesses:**

1. The proposed approach assumes a noise-free environment when adjusting the reward scale. However, if consider stochasticity—such as observation noise, reward corruption, or environmental uncertainty, the change of reward scale also affect these noise and may also have negative affect on the learning.

2. The hyperparameter study on soft update coefficent will strength the paper.

**Questions:**

1. The reward inflation process is happened before saved into replay buffer or after sample from replay buffer? if happened before replay buffer, then this will cause the issue that reward sclae uneven in the replay buffer which may limit the performance.

2.Is there a upper bound for the inflation which cause learning collapse (idea also from economics)?

---

> ### Author Response · Authors · 2025-12-03
>
> We thank the reviewer for acknowledging the novelty of our idea and the solid empirical results.
>
> We also sincerely appreciate the practical and insightful feedback provided.
>
> **Weakness 1. Stochasticity**
>
> We agree that examining the robustness of reward inflation in stochastic environments is a valid and crucial line of inquiry, particularly from a practical perspective. While robustness was not the initial scope, your suggestion prompted a valuable exploration into potential side effects.
>
> We interpreted the reviewer's suggestions into three noise types: **(1) observation noise, (2) reward noise, and action noise (environmental stochasticity)**.
>
> We constructed environments by adding a constant amount of Gaussian noise to MuJoCo tasks and compared the performance of no-inflation (0%) and 2% fixed-inflation agents. (Please see the newly added **Section 5.4, “Environmental Uncertainty”**.)
>
> The results, included in the revised manuscript (e.g., **Figure A.4**), show a performance drop for all agents in noisy environments, as expected. However, the 2% inflation agent consistently maintained performance comparable to, or slightly superior to, the no-inflation baseline across all three noise types. This new experimental evidence supports the conclusion that the reward inflation method retains its core benefits even when a moderate level of environmental stochasticity is introduced.
>
> **Weakness 2. FedeRL hyperparameter study ($\tau_\rho$)**
>
> We fully agree with the recurring concern regarding the tuning of FedeRL hyperparameters. We acknowledge that we overlooked providing adequate search studies or practical tuning guidance initially.
>
> We have addressed this by including a comprehensive hyperparameter search study in the revised manuscript (please see newly added **Section 5.4, "Hyperparameter Study of FedeRL"**).
>
> This study includes the soft update coefficient, $\tau_\rho$. Our analysis showed that while $\tau_\rho = 0.83$ (corresponding to an effective observation length of 6) demonstrated a slight edge, the overall conclusion is that FedeRL exhibits low sensitivity to most of its hyperparameters, reducing the tuning burden in practice.
>
> **Question 1. When does reward inflation happen?**
>
> This is the same important question raised by Reviewer #2CBW (Q3), and we have clarified this implementation detail in the revision.
>
> The reward inflation is applied before the experience is stored in the replay buffer. Thus, the reward scale in the replay buffer is maintained at the inflation level ($I(g)$) corresponding to the global step ($g$) at which the experience was collected.
>
> The reviewer's concern about the uneven reward scale in the buffer is valid.
>
> The confusion cost induced by reward inflation can be quantified using the replay buffer size $N$, with the maximum and average errors being $100\cdot((1+\tilde{\rho})^{N-1}-1)$% and $100\cdot\frac{2}{N^2} \sum_{k=0}^{N-1} (2k - N + 1) \cdot (1+\tilde{\rho})^k$%, respectively.
>
> For instance, when $N=10^6$ and $\rho_Y=0.02$ for $Y=10^5$, the maximum and mean errors are approximately $21.9$% and $7.3$%, respectively.
>
> However, this uneven scale is precisely the mechanism that forms the recency bias, one of the core effects of reward inflation. The time difference in rewards creates a disparity in TD errors, causing the agent to prioritize recent, higher-value experiences. Applying the inflation level at the sampling time (instead of the collection time) would eliminate this crucial recency bias effect (please refer to **Assumption 1**, **Theorem 1**, and **Remark 1** in the theoretical analysis, **Section 4.1**).
>
> **Question 2. Upper Bound for the Inflation**
>
> This is an interesting and highly reasonable suggestion. As in real-world economics where hyperinflation is avoided, and our own experiments where excessively high fixed inflation rates 10% showed adverse effects, setting an upper bound is a sensible safety measure.
>
> We agree that an upper bound is particularly beneficial for FedeRL as a safeguard against setting an overly aggressive instantaneous inflation rate. We consider this for future work, accordingly with the avoidance of exponential explosion mentioned by reviewer Fq7A.

---

### Official Review · Reviewer_Fq7A · 2025-10-30

**Soundness:** 3
**Presentation:** 4
**Contribution:** 2
**Rating:** 4
**Confidence:** 3

**Summary:**

This paper introduces "reward inflation," a novel setup for reward rescaling during RL inspired by monetary economics, where a 'nominal' reward scale is gradually incremented by some percent during training (inflation if a positive percent, deflation if a negative percent). The authors propose that when acting as an inflation, this reward modulation acts as an "incentive stimulus" that amplifies policy gradients, induces a recency bias in learning, and increases network activation (i.e. reducing the percent of dead neurons). They also propose FedeRL, an adaptive mechanism for setting the reward inflation percent with an analogy to a central bank which attempts to dynamically adjust inflation based upon an economic environment's speed of change. In this RL application, FedeRL dynamically adjusts the inflation rate based on two exponential moving averages of the policy gradient norms to stabilize training. Empirically, both fixed inflation and the FedeRL adaptive inflation method are shown to outperform the standard SAC baseline and other related methods on continuous control tasks.

**Strengths:**

- This work presents a highly original and creative paradigm by drawing a compelling analogy from monetary economics to RL. I genuinely found this interesting to read and consider.
- The adaptive FedeRL inflation rate determination is quite practical and shows clear, intuitive, and interpretable behavior.
- Consistent empirical performance gains are shown over the standard (0%) baseline and other comparable methods (like LR scheduling and PER).

**Weaknesses:**

- The primary theoretical effect (gradient norm amplification) is described as functionally very similar to learning rate scheduling, and yet the paper doesn't fully disentangle why "reward inflation" is a superior mechanism. When comparing against learning rate scheduling, this is only done to a limited degree (+ vs - 2%) when a more complete range of searching is completed for FedeRL and static inflation rates.
- The new FedeRL controller introduces its own set of hyperparameters (natural rate, short and long range moving norm averages, etc.), which may be difficult to tune and it is clear from Table 1 that these values might require tuning per task. These limitations are not explored.
- The inflationary rate is motivated heavily from a monetary perspective, though it is in the end a rather simple change to reward rescaling and any negative side effects (or limitations) of such are not much explored. E.g. catastrophic forgetting could be made much worse by any positive inflationary value. Furthermore, it is unclear whether the training time for models with inflationary rates which are non-zero has to be limited to avoid the eventual exponential rise in training impact. How the model training times must be limited is not discussed herein.
- This method is only applied with the SAC RL algorithm. Ensuring that these benefits transfer to alternative RL algorithms would be ideal and important for demonstrating generality.

**Questions:**

A response to the above weaknesses would be much appreciated.

Though the inflationary process is described as similar to learning rate scheduling, it might be more accurate to say that it could be similar to (perhaps even equivalent to) a particular scheduler for the learning rate combined with a change to the discounting term (gamma). Is this correct?

Ideally to address the above, the nominal return (sum of discounted future rewards) could be unpacked and given a form with respect to the regular reward. This should allow an identification of the degree to which these inflationary measures are simply equivalent to a modification of the discounting factor along with a learning rate change. Doing so could significantly demystify this method and also ensure that claims are not overblown. Note that this would also indicate that the baselines for comparison should also be extended with a sweep over learning rate scheduler rates and discounting factors.

---

> ### Author Response · Authors · 2025-12-03
>
> We sincerely thank the reviewer for recognizing the novelty and practicality of our work, as well as our efforts to present clear and consistent empirical results.
>
> We also appreciate the thoughtful feedback and agree that the concerns raised could not be fully addressed by the initial manuscript alone.
>
> **Weakness 1. Learning rate scheduling**
>
> We appreciate the reviewer’s insightful comment regarding the comparison between reward inflation and learning rate (LR) scheduling.
>
> The reviewer pointed out that if both reward inflation and LR scheduling share the same effect of amplifying the gradient norm, our explanation and experiments regarding the superiority of reward inflation appear insufficient.
>
> While this stems from a misunderstanding of our original intent, we fully acknowledge that this confusion arose because our descriptions were vague and not sufficiently comprehensive (originally in **Section 1, paragraph 4** and **Section 5.3, paragraph 4**).
>
> Technically, the learning rate ($\eta$) determines the magnitude of the update step applied to the weights ($\theta \leftarrow \theta + \eta \cdot \nabla \mathcal{L}$) and does not alter the gradient norm ($|\nabla \mathcal{L}|$) itself.
>
> However, by stating that "LR inflation is similar to reward inflation in terms of amplifying the learning signal" and by using the term "signal" polysemously throughout the paper, we inadvertently misled readers into believing that "LR inflation amplifies the gradient norm itself."
>
> We agree this is a critical distinction. We have corrected the manuscript to avoid using the term "signal" when describing LR inflation. Instead, we now explicitly clarify it as "reflecting the gradient more strongly during updates."
>
> Revisiting the two concepts with this clarification:
> - Reward Inflation: Amplifies the gradient norm.
> - LR Inflation: Increases the step size of the update based on the gradient.
>
> While these may appear similar at a glance, they are separate mechanisms that are effectively orthogonal. We originally included LR inflation as a baseline specifically to experimentally demonstrate this "seemingly similar but fundamentally different" nature. We have revised the text to make the intent and explanation of these experiments much clearer.
>
> Furthermore, the reviewer noted that while reward inflation was tested across various inflation rates, LR inflation was only tested at fixed rates of $\pm 2\%$, potentially weakening the objectivity of the comparison.
>
> We agree with this assessment. Following the reviewer's suggestion, we conducted additional experiments with LR inflation rates ranging from -4% to 4% and included these results in the revised manuscript (please see the newly added **Section 5.4, paragraph "Learning Rate Inflation"**).
>
> Interpreting the results based on our clarified intent:
> - Reward Inflation: Positive inflation led to performance improvements.
> - LR Inflation: Positive inflation degraded performance.
>
> These opposing results empirically demonstrate that the two effects are not equivalent.
>
> Meanwhile, we observed that LR deflation improved performance. However, this is effectively identical to **exponential LR decay**, a standard technique in machine learning, and yields improvements through a mechanism distinct from that of reward inflation.
>
> **Weakness 2. FedeRL hyperparameters**
>
> The reviewer noted that despite FedeRL's numerous hyperparameters, only a single configuration was used, and the potential challenges associated with tuning were not addressed.
>
> We fully agree with the reviewer’s comment and acknowledge that we initially overlooked providing an appropriate tuning guide or a parameter search study.
>
> To address this concern, we have provided a detailed description of each hyperparameter's characteristics and conducted a comprehensive hyperparameter search study in the revised manuscript. (Please refer to the newly added **Section 5.4, “Hyperparameter study of FedeRL”**.)
>
> Our analysis indicates that FedeRL exhibits relatively low sensitivity to hyperparameter variations.

---

> ### Author Response · Authors · 2025-12-03
>
> **Weakness 3. Limitations (side effects)**
>
> We thank the reviewer for pointing out the specific side effects of reward inflation. We fully agree that the reviewer’s examples (catastrophic forgetting and exponential explosion) are more concrete than our original description of 'inflation adaptation burden' or 'gradual reward noise.'
>
> We have addressed these two points as follows:
>
> - **Catastrophic forgetting**:
>
> The reviewer expressed concern that positive inflation might trigger catastrophic forgetting by diluting past data. While we agree with the mechanism, we view this as an intentional feature akin to 'creative destruction,' rather than a flaw.
>
> In the context of RL, the goal is not merely to optimize fixed historical data but to discover better policies through continuous exploration. Therefore, we prioritized 'plasticity' (acquiring new behaviors) over 'stability' (retaining old information), accepting the depreciation of past experiences to accelerate the learning of improved policies. This aligns with the well-known plasticity-stability dilemma in machine learning.
>
> However, we acknowledge that the manuscript lacked a quantitative explanation of how much stability is sacrificed. To clarify this, we analyzed the reward error based on memory size and inflation rate.
>
> The confusion cost induced by reward inflation can be quantified using the replay buffer size $N$, with the maximum and average errors being $100\cdot((1+\tilde{\rho})^{N-1}-1)$% and $100\cdot\frac{2}{N^2} \sum_{k=0}^{N-1} (2k - N + 1) \cdot (1+\tilde{\rho})^k$%, respectively.
>
> For instance, when $N=10^6$ and $\rho_Y=0.02$ for $Y=10^5$, the maximum and mean errors are approximately $21.9$% and $7.3$%, respectively.
>
> - **Exponential explosion**:
>
> We agree that unchecked reward inflation leads to exponential explosion, which poses a numerical limit to the model. We recognize this as a clear limitation.
>
> As the reviewer noted, this mirrors real-world economies where hyperinflation often necessitates currency reform (e.g., redenomination). While implementing such complex mechanisms is beyond the scope of this paper, we have explicitly added this to the **Discussion** section in the revision. We also noted that simple techniques, such as annealing the inflation rate to converge to 0% over time, can be effective practical solutions to prevent value explosion.
>
> **Weakness 4. Generality**
>
> The reviewer raised a valid concern regarding generalizability, noting that our experiments were exclusively conducted using the Soft Actor-Critic (SAC) algorithm.
>
> We fully agree with this point. While we initially maintained a single algorithm to ensure consistency across our diverse experiments, we recognize that relying solely on SAC limits the scope to continuous control domains and may obscure the method's wider applicability.
>
> To address this and enhance generalizability, we conducted new experiments applying reward inflation to DQN agents in the Atari domain, which utilizes a discrete action space. These results have been incorporated into the revised manuscript. (Please see the newly added **Section 5.4, “Arcade Learning Environment”**.)
>
> The results are compelling: in 26 out of 30 Atari 2600 games tested, the agent with a 2% inflation rate outperformed the baseline (0% inflation). This provides additional empirical evidence confirming the effectiveness of reward inflation across different algorithms and domains.
>
> **Question. Lr + gamma scheduling**
>
> We thank the reviewer for providing this thoughful suggestion for demystification of our method. We interpret the reviewer’s comment as a suggestion to analyze whether a scheduling method that simultaneously adjusts the learning rate and the discount factor ($\gamma$) could be mathematically equivalent to reward inflation.
>
> However, as discussed in our response to **Weakness 1**, while temporal adjustments to the learning rate and reward scaling may appear superficially similar, they play fundamentally different theoretical roles. Furthermore, regarding $\gamma$, it influences value estimation across steps within an episode (intra-episode horizon), whereas reward inflation operates based on global training steps. Therefore, their mechanisms differ significantly in nature.
>
> Nevertheless, we agree that a rigorous theoretical analysis of mathematical equivalence would be valuable for demystifying the underlying mechanisms. We plan to conduct such an indepth analysis in future work.

---

### Official Review · Reviewer_t5eG · 2025-11-01

**Soundness:** 1
**Presentation:** 2
**Contribution:** 1
**Rating:** 0
**Confidence:** 4

**Summary:**

This paper proposes a "reward inflation" paradigm inspired by monetary economics where the "nominal reward" scale is gradually increased during training, and introduces FedeRL, an adaptive controller mimicking the Federal Reserve's monetary policy. The authors analyze three effects of this approach: inducing recency bias, amplifying gradient norms, and enhancing neural activation. Empirical results on MuJoCo tasks are shown.

**Strengths:**

No

**Weaknesses:**

The paper's entire premise hinges on an analogy between RL rewards and money. This analogy is incorrect and unreasonable. It mistakes a simple heuristic (non-stationary reward scaling) for a economic principle.

**Questions:**

No

---

> ### Author Response · Authors · 2025-12-04
>
> We understand the reviewer's possible concerns regarding the analogies in this paper.
>
> However, drawing insights from other disciplines to broaden scientific perspectives is a well-established practice in academic research. Our intention was simply to explore novel viewpoints and methodologies in RL to add value, without contradicting or misrepresenting fundamental economic principles.
>
> We certainly acknowledge that there may be room for improvement in our interpretation of existing concepts or the clarity of our exposition.
>
> Regrettably, however, since the comment does not provide specific technical reasoning or concrete evidence, we are unable to offer a response that incorporates such feedback. We hope you understand our position.

---

### Meta-Review · Area_Chair_rhHA · 2025-12-11

**Summary:**

The paper introduces "Reward Inflation" and "FedeRL" drawing inspiration from monetary economics. The method gradually scales rewards during training to act as an incentive stimulus for policy learning.

While the economic analogy is novel, the paper appears limited by insufficient validation. Reviewers found the theoretical connection superficial and raised critical concerns regarding the method's stability (in particular value explosion) and its distinction from existing heuristics like learning rate scheduling.

While the authors provided a good rebuttal and expanded the manuscipt, including new experiments on Atari games and robustness checks against environmental noise. However, the paper requires additional work to delimit the validity of the method and its limitations. Specifically, it is unclear the limits of inflation to prevent divergence and a formal theoretical analysis (distinguishing the method from "learning rate + discount factor" scheduling) is missing.

On this basis, I cannot recommend this paper for acceptance.

**Reviewer Concerns:**

- The theoretical and empirical distinction between the proposed method and learning rate scheduling is not fully clarified (Fq7A).
- FedeRL introduces numerous hyperparameters (e.g., moving averages, natural rates) that need to be tuned without providing sufficient guidelines (Fq7A, Qmnv, 2CBW).
- It is unclear how the method impacts algorithm stability, in particular the potential value explosion and catastrophic forgetting (Fq7A, Qmnv).

The review from t5eG has been ignored as not constructive and poorly substantiated.

**Reviewer Scores:**

- Reviewer t5eG (Score: 0): This review was non-constructive and unsubstantiated, failing to engage with the authors. Consequently, this assessment has been disregarded for the meta-review.
- Reviewers Fq7A, Qmnv, 2CBW (Score: 4): These reviewers raised consistent technical concerns, which the authors addressed through extensive new experiments (e.g., Atari, noise robustness). While these improvements might have raised scores to marginal acceptance, the rebuttal was submitted late (most likely to accommodate the experimental workload) and did not benefit from a discussion with the reviewers.

---

### Decision · Program_Chairs · 2026-01-26

Reject